# Identification of Responders to Balneotherapy among Adults over 60 Years of Age with Chronic Low Back Pain: A Pilot Study with Trajectory Model Analysis

**DOI:** 10.3390/ijerph192214669

**Published:** 2022-11-08

**Authors:** Benjamin Raud, Charlotte Lanhers, Cindy Crouzet, Bénédicte Eschalier, François Bougeard, Anna Goldstein, Bruno Pereira, Emmanuel Coudeyre

**Affiliations:** 1Service de Médecine Physique et de Réadaptation, CHU Clermont-Ferrand, INRAE, Université Clermont Auvergne, F-63000 Clermont-Ferrand, France; 2Département de Médecine Générale, Université Clermont Auvergne, F-63000 Clermont-Ferrand, France; 3Unité de Biostatistique, Délégation Recherche Clinique et Innovation, CHU Clermont-Ferrand, 63003 Clermont-Ferrand, France

**Keywords:** balneotherapy, chronic low back pain, aged, hydrotherapy, disability

## Abstract

Balneotherapy may be a relevant treatment for chronic low back pain (LBP) in individuals > 60 years old. This pilot study aimed to determine the effectiveness of balneotherapy for chronic LBP in people > 60 years old and to determine profiles of responders with trajectory model analysis. This was a pilot prospective open cohort study, with repeated measurements using validated questionnaires; participants were their own controls. The primary endpoint was the proportion of participants with a change in pain intensity between the start of treatment and 3 months after treatment assessed with a numeric scale (NS) from 0 to 100 mm, with an effect size (ES) > 0.5. The assessments involved questionnaires that were self-administered on days (D) 1 and 21 and at months 3 and 6. The secondary objective was to determine the profile of responders to balneotherapy. We included 78 patients (69.2% women), mean age 68.3 ± 5.3 years. The mean pain score on the NS was 48.8 ± 19.9 at D1 and 39.1 ± 20.5 at 3 months (*p* < 0.001). The ES was 0.47 [95% confidence interval [CI] 0.25 to 0.69] for the whole sample; 36% (28/78) had an ES > 0.5; 23% (18/78) had a moderate ES (0 to 0.5); and 41% (32/78) had an ES of zero (14/78) or < 0 (18/78), corresponding to increased pain intensity. The pain trajectory model showed that the change in pain between D1 and D21 for trajectory A (larger reduction in pain intensity) was −50% [95% CI −60 to −27], and for trajectory B (smaller reduction in pain intensity), it was −13% [−33 to 0] (*p* < 0.001). Between Day 1 and month 3, the change for trajectory A was −33% [−54; 0] and for trajectory B was −13% [−40 to 0] (*p* = 0.14). Finally, between D1 and month 6, the change for trajectory A was −50% [−60 to 0] and for trajectory B was −6% [−33 to 17] (*p* = 0.007). The patients in trajectory A reported performing more physical activity than those in trajectory B (*p* = 0.04). They were also less disabled, with a mean Oswestry Disability Index of 40.4 versus 45.7 for those in trajectory A and B, respectively, (*p* = 0.03) and had a higher total Arthritis Self-Efficacy Scale score. This real-life study of the effectiveness of balneotherapy on chronic LBP identified distinct pain trajectories and predictive variables for responders. These criteria could be used in decision-making regarding the prescription of balneotherapy, to ensure personalized management of chronic LBP.

## 1. Introduction

Chronic low back pain (LBP) is a major public health issue, that generates considerable direct and indirect socioeconomic costs [1,2]. Chronic LBP is estimated to account for more than 85% of the medical costs of all cases of LBP [3], although it represents only 10% to 15% of the pain [4]. The estimated cost of chronic LBP in France is 2.7 billion euros per year (i.e., 1.5% of the annual health expenditure [figure from the Public Health Orientation Law]). In the coming years, with the aging of the population, this burden is likely to increase; in 2050, 31.9% of the French population will be >60 years old [5,6]. Chronic low back pain can cause disability, have a negative impact on mood, and reduce health-related quality of life [2]

A wide range of treatments for chronic LBP exist, including pharmacological treatments, such as nonsteroidal anti-inflammatory drugs (NSAIDs), and non-pharmacological treatments, such as physiotherapy and adapted physical activities. In 2019, the new recommendations of the French National Authority for Health proposed a therapeutic strategy to manage chronic LBP [7]. The only treatment recommended with grade A evidence was NSAIDs. Physiotherapy and adapted physical activities were also recommended as first-line treatment (grade B). Paracetamol has been evaluated in a few studies [8], but none found that it effectively managed chronic LBP. Nevertheless, expert agreement recommends paracetamol, and it is cited as a first-line drug therapy. Although NSAIDs are recommended as a first-line treatment, they can have numerous adverse effects, particularly in older people [9,10]: their long-term use can affect the cardiovascular and gastrointestinal systems. Furthermore, the recommendations focused on non-specific LBP in individuals aged 18 to 55 years old. To our knowledge, there are no specific recommendations for the management of chronic LBP in individuals above this age.

We conducted a review on PubMed using the keywords “chronic low back pain” with the limitations “guideline, systematic review, ages + 65” for the last 10 years”. We did not find any guidelines but did find three reviews. The prevalence of LBP among elderly individuals ranged from 21 to 75%. [11]. Exercise is the most consistent intervention in optimizing recovery from LBP, but the review was unable to draw conclusions regarding the number and frequency of sessions that should be done [12]. Among the studies included in the systematic review, six are related to physiotherapy modalities; one is related to strength training, and two are related to endurance training. The low-quality evidence suggests that physical therapy is associated with a small-to-moderate reduction in pain and a small improvement in function. [13]

From a personalized medicine perspective, individuals’ preferences must be considered; this could be facilitated by a wider range of evidence-based treatment options.

In the 2000 recommendations [14], balneotherapy had grade B evidence, although it was not evaluated in the latest recommendations. Balneotherapy involves bathing in hot or warm baths of natural mineral water in spa centers (Mesh). Several clinical trials have demonstrated the effectiveness of balneotherapy on pain and functional disability in chronic LBP [15,16]. A study of individuals > 50 years old with chronic LBP, who had balneotherapy treatment for 15 days, showed a reduced consumption of analgesia, decreased pain levels, and improved functional capacity 3 months after the treatment [15].

People > 60 years old are classified as “elderly” by the World Health Organization (WHO). Back pain is quite prevalent in this population and has a significant impact on the daily lives of those affected [16,17,18]. The effectiveness of balneotherapy has not been specifically evaluated in patients > 60 years old.

The results of studies on balneotherapy for the treatment of LBP are contradictory. The 2004 European recommendations did not recommend the use of balneotherapy for the management of chronic LBP [19]. However, in 2015, a systematic review and meta-analysis found that pain intensity (visual analog scale [VAS]) and disability (Oswestry Disability Index [ODI]) reduced significantly more after the balneotherapy than the control treatment [20]. Further studies are therefore needed to determine if there is a particular profile of older individuals who respond to balneotherapy [21].

The aim of this study was to measure the effectiveness of conventional balneotherapy in older individuals with chronic LBP on pain intensity. The secondary aim was to determine the profile of good responders to balneotherapy with trajectory model analysis.

## 2. Materials and Methods

This was a pilot, single-center, prospective open cohort study, with repeated measurements using validated questionnaires (Appendix A). The study took place over 6 months; the participants were their own controls, and the individual post-treatment change was analyzed.

The study took place at the Neyrac-les-Bains thermal baths over 4 months. All the treatments were identical and pre-determined and were administered 6 days per week for 3 weeks. Four of the following hydrotherapy treatments were administered each day: thermal steam bath at 45–48 °C, cataplasm, aerobath with mud, general jet shower, mobilization pool, massage under thermal water, application of thermal water gel, exercise in hot thermal mud, and high-pressure shower. The whole treatment lasted for 2 h.

The composition of the water was as follows (per 1 L): silica 127.6 mg anions, H_2_CO_3_ 1711.1 mg, SO_4_ 16.6 mg, Cl 10.4 mg, cations, Ca++ 217 mg, Mg++ 63 mg, Na+ 274.4 mg, B 270 mg, Cu < 0.02 µg, Ph 6.45, Zn < 0.02 µg, Fe 15 mg, Se < 5 µg.

To be included in the study, individuals had to be aged 60 to 80 years old and have had mechanical lumbar spine pain for >3 months. They were not included if they had a history of spinal surgery, inflammatory rheumatic disease, a tumor or a traumatic or infectious lesion, received balneotherapy in the 6 months before the start of the study, an evolving disc hernia, a numeric scale (NS) rating of 0 at inclusion, difficulty understanding written or oral French, or were unable to participate in the study.

### 2.1. Endpoints

The primary endpoint was the proportion of participants with a change in pain intensity on the NS between the start of the treatment and 3 months post-treatment with an effect size (ES) > 0.5. According to the recommendations proposed by Cohen [22]: ES < 0.2 was considered insignificant, 0.2 to 0.5 small, 0.5 to 0.8 moderate, and > 0.8 large.

The secondary endpoints were functional disability, measured with the Oswestry Disability Index (ODI) [23] (a 10-item questionnaire, the score is calculated as a percentage of disability from 0% to 100%); perceived self-efficacy, measured with the Arthritis Self-Efficacy Scale (ASES) (a questionnaire that is validated in French [24] and has three parts: six questions about personal pain management, nine questions about function, and six questions about other symptoms); and fears and beliefs, measured with the physical part of the Fear Avoidance Belief Questionnaire (FABQ) (evaluates fears and beliefs about physical activity and pain) [25]. The consumption of analgesia was assessed by standardized collection (molecule and dosage). We did not use the work dimension of the FABQ, given the age of the target population.

We also determined the profile of good responders to balneotherapy with trajectory model analysis.

### 2.2. Data Collection

The evaluations involved self-administered questionnaires that were distributed personally at the end of the inclusion visit (day [D] 1) and at the end-of-treatment visit (D21), as well as being sent by mail to the participant’s home at 3 and 6 months. For the evaluations at D1 and D21, a nurse from the balneotherapy study was available to help participants complete the questionnaires, if necessary. The questionnaires were then collected from a box in the entrance hall of the Neyrac-les-Bains spa. For the 3- and 6-month evaluations, the investigator contacted the participants by telephone in the case of non-return.

### 2.3. Statistical Analysis

We calculated the number of participants required according to Fleming’s multistage design [26] because this was a pilot study that evaluated the ES associated with the benefits of conventional balneotherapy in older individuals with chronic LBP. This experimental design considered a single group, and the sequential method analyzed efficacy, set as a dichotomous criterion (and, more precisely, its confidence interval [CI]): the proportion of participants with a reduction in pain level with an ES > 0.5 at month 3. In view of the recruitment capacity, feasibility, and literature data, we chose a one-stage design with a lower bound of maximum inefficiency of about 50% and an upper bound of 65% (minimum efficiency), for the alpha and beta error risks set at 0.05 and 0.20, respectively (power of at least 80%). With these assumptions, we needed to include 78 individuals. We considered that a reduction in pain intensity with an ES > 0.5 for ≥47/78 individuals would demonstrate that balneotherapy was effective.

Continuous data are expressed as the mean ± standard deviation, or the median and [interquartile range], depending on the normality of their distribution (assessed with the Shapiro–Wilk test). To analyze the longitudinal data, random-effects models for repeated data were used, with time as a fixed effect and the individual as a random effect, to take between- and within-individual variability into account. A Sidak’s Type I error correction was applied for multiple comparisons. For continuous endpoints, the normality of residuals was evaluated with the Shapiro–Wilk test. When appropriate, a logarithmic transformation was proposed to achieve the normality of the dependent outcome. The results are expressed as ESs and 95% CIs and interpreted according to Cohen’s recommendations. To identify the distinct trajectories of pain, we used semi-parametric mixture models (group-based trajectory model) to model the relation between pain level and time, for each trajectory, the shape of the trajectory, and the estimated proportion of the population belonging to each trajectory. This model allows data to be grouped using different parameter values for each group distribution. The groupings may identify distinct subpopulations. The analysis provides a formal way to determine the best-fit number of trajectories and a precise estimate of group membership allocation, which can be expressed by using observed probabilities and posterior probabilities. Furthermore, the best-fitting model was selected according to the Bayesian Information Criterion. The continuous variables were then compared between the independent trajectory groups with the Student’s t-test or Mann–Whitney test. Homoscedasticity was analyzed with the Fisher–Snedecor test. The categorical variables were compared with the Chi-square or Fisher’s exact test. The statistical analyses were performed with Stata 15 (StataCorp, College Station, TX, USA). All the tests were two-sided and *p* < 0.05 was considered statistically significant.

### 2.4. Legal and Ethical Aspects

This project received ethical approval from the Comité de Protection des Personnes Sud Est VI on 26 April 2016 (2016/CE 28) and was registered on ClinicalTrials.gov (NCT02894125). The individuals could participate in this study only after receiving verbal information from the doctor. All the information was summarized in a written information note given to each person. At any time, the participants could exercise their right of access and rectification by contacting the study managers. Written consent was obtained and kept. The anonymity of the participants was ensured by an anonymous number on all the documents necessary for the research. All the research was conducted in accordance with the Declaration of Helsinki.

## 3. Results

### 3.1. Participants

We included 78 individuals who met the inclusion criteria; 69.2% were women, the mean age was 68.3 ± 5.3 years, and the mean weight was 72.8 ± 15.5 kg. The data were analyzed at D1, D21 (end of treatment), and at 3 months for all 78 participants (Figure 1). One participant was lost for follow-up at 6 months due to hospitalization for another reason. All the individuals were referred by their physician who made the diagnosis; they were then seen by a specialist in the spa resort who confirmed the diagnosis. The diagnosis of chronic LBP was made following a clinical examination under the HAS guidelines, as low back pain is defined by pain between the costal margins and the inferior gluteal folds. In the absence of any red flags, no imagery was required.

### 3.2. Analysis of Effectiveness of Balneotherapy on Pain

The mean pain intensity NS rating was 48.8 ±19.9 points on D1 and 39.1 ± 20.5 at month 3 (*p* < 0.001, ES = 0.47 [95% CI 0.25 to 0.69]). In total, 28/78 (36%) participants had an ES > 0.5; 18/78 (23%) had a moderate ES (0 to 0.5); and 32/78 (41%) had no effect size (14/78) or an ES < 0 (18/78) (i.e., an increase in pain intensity).

The following variables had no effect on the change in pain: age (*p* = 0.87, mean age 67.5 ± 4.8 vs. 68.3 ± 5.3 for all participants with an ES > 0.5), sex (*p* = 0.50, 46% of men vs. 41% of women with ES > 0.5), weight (*p* = 0.90, mean 73.8 ± 17.7 kg vs. 72.8 ± 15.5 for all participants with ES > 0.5), family situation (*p* = 0.36), level of physical activity (*p* = 0.33), and the number of previous spa treatments in the center (*p* = 0.74). On D21, the mean pain level was 35.5 ± 19.5 (*p* < 0.001, ES = 0.64 [95% CI 0.42 to 0.86]) and, at month 6, was 39.7 ± 22.3 (*p* < 0.001, ES = 0.44 [0.22 to 0.66]).

### 3.3. Effect of Balneotherapy on ODI, ASES, and FABQ

Significant changes were found in ODI, ASES, and FABQ over time; these results are presented in Table 1.

### 3.4. Trajectories of Pain

The trajectory model approach identified two distinct pain profiles (Figure 2): trajectory A: a larger reduction in pain intensity, 41.0% of participants; and trajectory B: a smaller reduction in pain intensity, 59% of trajectories (observed probabilities). The posterior probabilities were 42.8% and 57.2%, respectively, (expected close to the aforementioned observed probabilities); the average posterior probabilities were 94.7% and 93.2% (expected >70%); and the odds of correct classification based on the posterior probabilities of group membership 25.8 and 9.6 (expected >5).

The change in pain level from D1 to D21 was −50% [95% CI −60 to −27] for trajectory A and −13% [−33 to 0] for trajectory B (*p* < 0.001). The change from D1 to month 3 was −33% [−54 to 0] for trajectory A and −13% [−40 to 0] for trajectory B (*p* = 0.14). Finally, the change from D1 to M6 was −50% [−60 to 0] for trajectory A and −6% [−33 to 17] for trajectory B (*p* = 0.007). Comparisons of the participant characteristics between the two pain trajectories are provided in Table 2. The participants were similar in terms of weight and age, but those with trajectory A declared doing more physical activity than those with trajectory B: 46.9% performed > 3 h per week of physical activity versus 19.7% respectively (*p* = 0.04) (Table 3). Those with trajectory A also had fewer functional limitations than those in trajectory B (mean ODI 40.4 vs. 45.7; *p* = 0.03) and a higher total ASES score as well as higher ASES subscores (all *p* < 0.05).

## 4. Discussion

Three weeks of balneotherapy for chronic LBP resulted in a mean reduction of pain intensity of 10 mm on the NS, with an ES of 0.47. The pain intensity was reduced with an ES > 0.5 for only 36% (28/78) of the participants. Furthermore, the pain intensity was higher at month 3 than before treatment for 23% and did not change for 18% of the participants.

We identified two different pain trajectories with a trajectory model analysis. The first trajectory (A) involved a clear decrease in pain intensity (50% decrease) immediately after the treatment (D21), with the effects continuing until month 3 (33% decrease from baseline) and month 6 (50% decrease). For the second trajectory (B), the pain intensity reduced, to a lesser extent, immediately post-treatment (13% reduction) and gradually returned to the pre-treatment state, with a 13% and 6% improvement at months 3 and 6, respectively. The mean ODI score for the whole sample reduced from 43.5 to 39.5, with an ES of 0.48 at month 3, indicating a reduction in the disability level.

To our knowledge, this is one of the few studies to focus on chronic LBP in individuals > 60 years old [27]. The characteristics of our sample were similar to those of people usually treated in a balneotherapy setting for chronic LBP [20].

Our results showed a positive effect of balneotherapy on pain, as has been found previously [21], with a moderate ES of 0.47. A meta-analysis conducted in 2007 listed the ESs of the different therapies available for chronic LBP [28]. None of the ES were much higher than that found for the balneotherapy in the present study: 0.25 for physical activity (0.52 in the long term), 0.57 for cognitive-behavioral therapies (0.24 in the long term), 0.22 for transcutaneous electrical nerve stimulation, and 0.35 for spinal manipulation. Only NSAIDs had an ES of 0.61 (relative risk) in the short term. However, balneotherapy has the advantage of being safe [29,30], especially in older people in whom complications relating to NSAID-use are frequent [31,32].

We considered that an improvement in >50% of the participants would indicate that the balneotherapy was effective; however, improvement only occurred in 36%. The pain trajectory analysis showed that a proportion of the participants had a clear improvement by D21, with the benefits on pain persisting for up to 6 months. It also showed that, despite the apparent good average reduction in pain (−10 mm for the whole population), little, or no, improvement occurred in a proportion of participants (trajectory B). These results indicate that some individuals respond to balneotherapy whereas others do not.

Our purpose was to identify the criteria that could predict response to balneotherapy; we were unable to discern a typical profile, although we found several predictive variables. As shown in Table 1, the responders had less initial disability, more frequently consumed strong analgesics (stage II), had higher levels of self-efficacy, and were more physically active than non-responders. The evolution of the participants in trajectory B shows that some individuals showed an altered condition, whether function or pain, after the balneotherapy.

These results suggest that balneotherapy alone is not sufficient for the treatment of chronic LBP and, if used, it should be associated with a global management program, including specific exercises and/or an educational program. Programs that combine specific treatment with balneotherapy, 6 days per week for 3 weeks with following hydrotherapy treatments, thermal steam bath at 45–48 °C, cataplasm, aerobath with mud, general jet shower, mobilization pool, massage under thermal water, application of thermal water gel, exercise in hot thermal mud, and high-pressure shower could reduce pain and disability in individuals identified as responders [33].

### Limitations

The aim of this study was to identify pain trajectory profiles; therefore, the open-label design with no control or placebo group was not a limitation. However, although observational research is valuable in bringing information needed to improve medical decision-making, it is associated with some bias.

Another limitation is that the follow-up only lasted 6 months and not 12; this is the main limitation of our study, but the effect of balneotherapy is at the medium term and no study had ever demonstrated a long-term effect.

Either, we did not consider several psychosocial factors that could affect the experience of chronic pain, particularly mood.

To finish with, we did not include a large sample size because we considered that a reduction in pain intensity with an ES > 0.5 for ≥47/78 individuals would demonstrate that balneotherapy was effective. Also, for this pilot study, we did not have any comparative group and only included participants older than 60 years, because that was the aim of the study. Another study with all ages would be relevant.

## 5. Conclusions

Despite numerous limitations, this real-life study of the effectiveness of balneotherapy on aged participants with chronic LBP confirmed a medium-term effect at 3 months and identified distinct pain trajectories and predictive variables for responders. Further studies are required to fully determine the subgroups of individuals over the age of 60 who benefit the most from balneotherapy. The ES for pain after the balneotherapy was similar to that for other recommended therapies for chronic LBP. New studies must take into account a long-term follow-up for this chronic pathology, a larger sample size and a comparative group, and also participants of other age groups.

Despite the apparent efficacy of the treatment, we identified two responder profiles, with some individuals clearly responding less to the balneotherapy than others. The criteria for responders identified in this study could be used in decision-making regarding the prescription of balneotherapy to ensure personalized management of chronic LBP.

## Figures and Tables

**Figure 1 ijerph-19-14669-f001:**
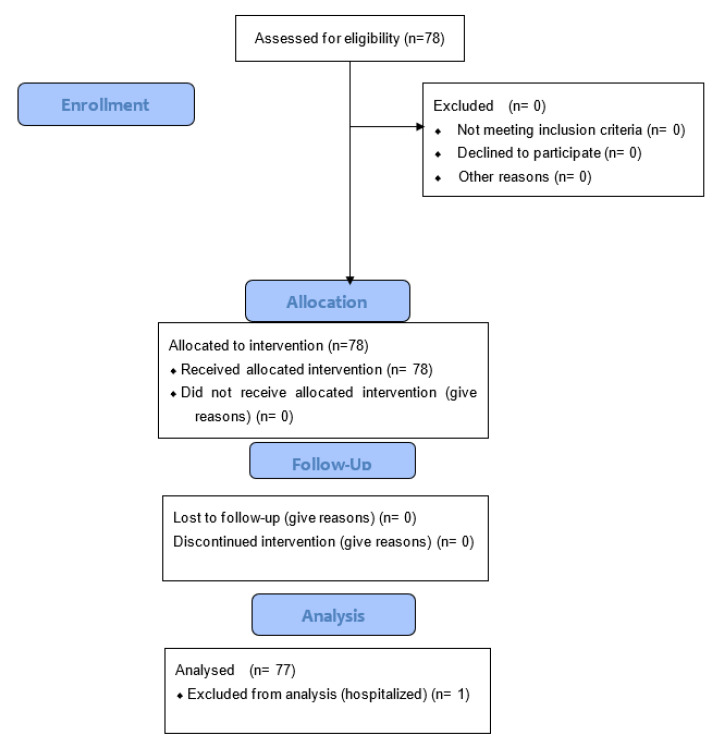
Flow diagram.

**Figure 2 ijerph-19-14669-f002:**
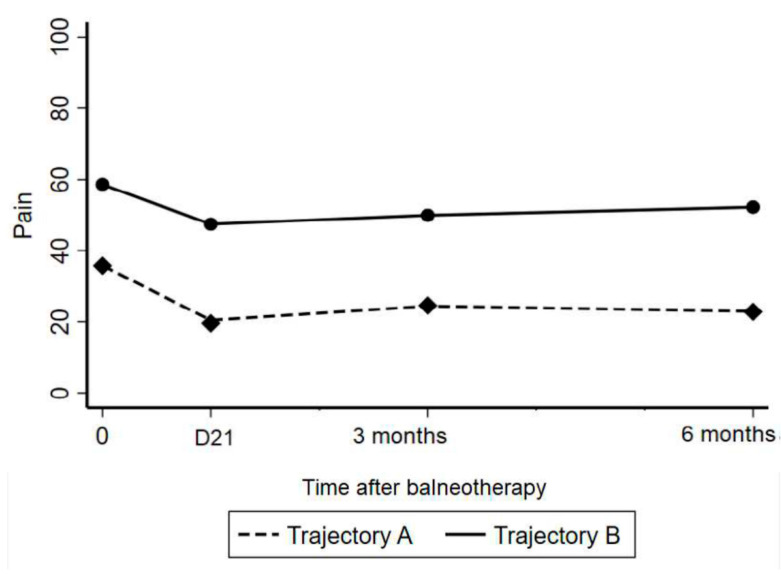
Change in pain intensity from the start of balneotherapy until the 6-month follow-up for both pain trajectories.

**Table 1 ijerph-19-14669-t001:** Change in Oswestry Disability Index, ASES, and FABQ scores over time.

	D1	D21	Month 3	Month 6
ODI	43.5 ± 10.6	40.4 ± 8.7 *0.37 [0.15 to 0.59], p = 0.001*	39.4 ± 9.5 *0.48 [0.26 to 0.71], p < 0.001*	40.3 ± 10.3 *0.37 [0.14 to 0.59], p = 0.001*
ASES: pain	33.2 ± 8.9	35.2 ± 8.2 *0.23 [0.01 to 0.45], p = 0.045*	34.3 ± 9.2 *0.13 [−0.09 to 0.35], p = 0.26*	33.8 ± 9.8 *0.07 [−0.16 to 0.29], p = 0.56*
ASES: function	71.0 ± 15.6	72.4 ± 13.9 *0.11 [−0.11 to 0.33], p = 0.33*	71.2 ± 14.3 *0.02 [−0.21 to 0.24], p = 0.89*	72.1 ± 16.5 *0.08 [−0.14 to 0.31], p = 0.46*
ASES: other symptoms	47.7 ± 8.4	45.2 ± 10.7 *0.28 [0.06 to 0.50], p = 0.01*	47.1 ± 9.5 *0.22 [0.00 to 0.44], p = 0.05*	47.2 ± 10.2 *0.23 [0.01 to 0.45], p = 0.04*
FABQ	18.3 ± 5.3	18.7 ± 4.5 *0.07 [−0.15 to 0.29], p = 0.53*	18.2 ±5.1 *−0.02 [−0.25 to 0.20], p = 0.83*	18.5 ± 5.5 *0.04 [−0.18 to 0.27], p = 0.69*

Data are mean ± standard-deviation and effect-size with 95% confidence interval and *p*-value (in italics) for comparisons with D1 values. ODI: Oswestry Disability Index, ASES: Arthritis Self-Efficacy Scale; FABQ: Fear Avoidance Beliefs Questionnaire.

**Table 2 ijerph-19-14669-t002:** Pain trajectories (*n* = 77).

	Trajectory A	Trajectory B	*p*-Value
Age (year)	68.1 ± 4.9	68.5 ±5.6	0.77
Weight (kg)	74.9 ± 14.4	71.4 ± 16.2	0.33
Physical activity			0.04
<1 h/week	4 (12.5%)	9 (19.6%)	
1–3 h/week	13 (40.6%)	28 (60.9%)	
>3 h/week	15 (46.9%)	9 (19.7%)	
Number of spa treatments			0.14
1	4 (12.5%)	14 (30.4%)	
1–5	17 (53.1%)	14 (30.4%)	
≥6	11 (34.4%)	18 (39.1%)	
Physiotherapy	9 (28.1%)	15 (32.6%)	0.67
NSAIDs	7 (21.9%)	16 (34.8%)	0.22
Paracetamol	20 (62.5%)	36 (78.3%)	0.13
WHO level II pain killer	3 (9.4%)	16 (34.8%)	0.02
WHO level III pain killer	0	0	
ODI (0–100%)	40.4 ± 10.2	45.7 ± 10.5	0.03
ASES score			
Pain (0–50)	35.6 ± 6.8	31.5 ± 9.8	0.03
Function (0–90)	76.3 ± 12.3	67.3 ± 16.6	0.008
Other symptoms (0–60)	48.9 ± 8.4	42.6 ± 11.5	0.007
Total	160.8 ± 23.5	141.5 ± 32.4	0.003
FABQ physical score (0–24)	19 ± 4.7	17.8 ± 5.6	0.32

Data are mean ± SD unless otherwise indicated. ASES, Arthritis Self-Efficacy Scale; FABQ, Fear Avoidance Belief Questionnaire; NSAIDs, non-steroidal anti-inflammatory drugs; ODI, Oswestry Disability Index; WHO, World Health Organization.

**Table 3 ijerph-19-14669-t003:** Sample characteristics at baseline (*n* = 78).

		n	%
Physical activity	<1 h/week	13	16.7
	1–3 h/week	41	52.6
	>3 h/week	24	30.7
1st spa treatment	Yes	10	12.8
Number of previous spa treatments	1	18	23.1
	1–5	31	39.7
	≥6	29	37.2
Physiotherapy		24	30.8
NSAIDs		23	29.5
Paracetamol		56	71.8
WHO level II pain killer		19	24.4
WHO level III pain killer		0	0
ODI (0–100%) (mean ± SD)		43.5 ± 10.6	
ASES score	Pain (0–50)	33.2	
	Function (0–90)	71	
	Other symptoms (0–60)	45.2	
	Total	149.4	
FABQ score (0–24)	Physical	18.3	

ASES, Arthritis Self-Efficacy Scale; FABQ, Fear Avoidance Belief Questionnaire; NSAIDs, non-steroidal anti-inflammatory drugs; ODI, Oswestry Disability Index; WHO, World Health Organization.

## Data Availability

**The** data supporting the reported results can be requested from the authors.

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
