# Peer review of "Identification of Responders to Balneotherapy among Adults over 60 Years of Age with Chronic Low Back Pain: A Pilot Study with Trajectory Model Analysis"

_ijerph, 2022, doi:10.3390/ijerph192214669_

Round 1

Reviewer 1 Report

the present study is well written with clear and methodologically sound objectives. I have only a few minor revisions to propose to the authors. 

Introduction

I would appreciate in the introduction a more comprehensive paragraph on chronic low back pain encompassing also the impact on quality of life, mood, physical functioning. 

It might be useful in the introduction to emphasize the importance of also evaluating nonpharmacological treatments by pointing out the limitations of such treatment: disks of chronic use of NSAIDs (cardiovascular gastrointenstinal risks). In a personalized medicine perspective, the patient's preferences should also be taken into account; having a wider range of evidence-based treatment options is helpful in moving in this direction.

line 72: please add reference.

Discussion

line 247: typos

line 293: typos

I suggest adding in the discussion section that several psychosocial factors known to affect the experience of chronic pain were not taken into account. Mood, especially, could have been an interesting aspect to consider. 

Also, please add a paragraph addressing the limitations of the study.

Author Response

Introduction

I would appreciate in the introduction a more comprehensive paragraph on chronic low back pain encompassing also the impact on quality of life, mood, physical functioning. 

Thank you, we added a sentence in the introduction

It might be useful in the introduction to emphasize the importance of also evaluating nonpharmacological treatments by pointing out the limitations of such treatment: disks of chronic use of NSAIDs (cardiovascular gastrointenstinal risks).

We added some elements about the risks of NSAIDs associated with chronic use, particularly gastrointestinal and cardiovascular effect.

In a personalized medicine perspective, the patient's preferences should also be taken into account; having a wider range of evidence-based treatment options is helpful in moving in this direction.

We thank the reviewer for this remark and we modified the text accordingly.

line 72: please add reference.

We added a reference

Discussion

line 247: typos, done

line 293: typos, done

I suggest adding in the discussion section that several psychosocial factors known to affect the experience of chronic pain were not taken into account. Mood, especially, could have been an interesting aspect to consider. 

Also, please add a paragraph addressing the limitations of the study.

 We added a limitations paragraph and some elements about psychological factors

Reviewer 2 Report

Congratulations to the authors for this research work, but the manuscript requires a number of important changes and corrections. The authors of the article set the following objective: effectiveness with the benefit of spa treatment for chronic LBP in people > 60 years old and to determine profiles of responders with trajectory model analysis.

The article requires an in-depth review of the English language by an expert.

My comments and suggested changes to the article are as follows:

1º The keywords should be a search and identification tool for your research work, look for the words that most closely resemble your manuscript, it would be advisable to use MesH terms, since you have not indicated for example the term hydrotherapy and balneotherapy.

2º The title of the manuscript is unstructured and a bit confusing, rephrase the title, make it clear what type of study it is, is it a pilot study or an observational study, does it occur in patients with chronic pain or over 60 years old, can 30 year old patients have chronic low back pain?

Indicate type of study in title and abstract.

4º It should give a detailed and exhaustive description of the study population; incidence; classification; diagnosis; current treatments; physical, psychological, social and economic problems associated with the injury, according to the recommendations of the Guidelines for low back pain (an important aspect of the justification and importance of the study) both in the introduction and in the discussion.

5º You use the following reference to justify in the text the use of balneotherapy "In 2009, in a systematic review of randomized controlled trials, Falagas et al. showed the effects of balneotherapy on pain in chronic LBP in 25 trials and a beneficial but statistically insignificant effect in 8 trials. The duration of effectiveness on pain ranged from 10 80 days to 1 year" this reference is more than 13 years old, there are more current studies such as Bai R, Li C, Xiao Y, Sharma M, Zhang F, Zhao Y. Effectiveness of spa therapy for patients with chronic low back pain: An updated systematic review and meta-analysis. Medicine (Baltimore). 2019 Sep;98(37):e17092. doi: 10.1097/MD.00000000000000017092. The introduction should be rewritten with more current bibliographic references, some are more than 10 years old, therefore the study presented is not justified with current evidence, which potentially limits the results obtained.

6º The text is poorly written and gives rise to many doubts in the terms used, you refer to the treatment technique in different ways throughout the manuscript you call it Spa Treatment or Spa therapy or even allude to balneotherapy. What is spa therapy? In which similar populations has it been applied? What are the techniques that compose it? You should further develop this whole concept you are talking about and what has been done to date.

7º The work deals with chronic low back pain and why can only people over 60 years of age participate? what is the basis for this? what clinical guide advises that it should only be from 60 years of age? why do they exclude a younger population? can a 30 year old person have chronic low back pain? It is important to justify this aspect in the introduction and that it be reflected in the limitations of the study (there is no section on the limitations of the study in the discussion).

8º How were these participants diagnosed? Where were these participants diagnosed and referred from? Who performed the evaluation to determine that it was chronic low back pain?

9º They should indicate the code provided by the ethics committee that approved this study with humans and that it is reflected that it complies with the declaration of Helsinki.

10º The sample size is very small. Why is the number of participants so small? How did you solve this problem? How was the sample size determined?

11º If the pathology under study is chronic, why was a long-term follow-up of the variables analyzed not performed?

12º They should insert the section on material and methods: study design, participants, outcomes, sample size, interventions (more detail on the techniques used, time spent in each of them and duration of the sessions, number of sessions per week). There were dropouts of participants, for reasons.

13º It would be interesting to insert a flow diagram to better understand the process and images of the therapies used in the intervention.

14º Divide the sections of the results explaining each of the variables analyzed, the first part of the results are the characteristics of the participants.

15º Discussions should cover the key findings of the study: discuss any previous research related to the topic to place the novelty of the discovery in the appropriate context, discuss possible shortcomings and limitations in its interpretations, discuss its integration into the current understanding of the problem and how. This advances current views, speculates on the future direction of research, and freely postulates theories that could be tested in the future, completed, and reformulated. The discussion should be rewritten to present serious errors.

There is no justification with the current literature of their results both for and against, as there are current studies of the use of balneotherapy, spa therapy in patients with chronic low back pain which have not been taken into consideration to discuss the results, I advise you to rewrite the discussion, then I attach a search where you have clinical trials and reviews on the topic of study that can be complementary to your discussion: https://pubmed.ncbi.nlm.nih.gov/?term=balneotherapy+and+low+back+pain&sort=date

16º It presents very outdated and old references which do not correspond to the current evidence, reformulate its introduction and discussion according to the current literature and evidence, see references with more than 12 years of antiquity regarding the subject: [1,2,6,8, 9,10,13,14,15,16,17,19,21,24,26,27,28,32]

17º The bibliography is not adapted to the standards of the journal.

The citations of the references in the text are erroneous, an in-depth revision of this aspect is needed, I recommend the authors to review the author's guide of the journal. Use the reference style, based on the American Chemical Society (ACS) style, both in the text and in the bibliography section, as indicated in the author's guide of the journal.

18º You must reformulate your conclusions, referring only to the results obtained by you and mentioning the obtaining of conclusions with caution given the high number of limitations that this research project presents, such as the non-long-term follow-up of a chronic pathology, the sample size, no comparative group, the dispersion of treatment techniques, the selection bias of participants only older than 60 years, which leaves an important group of the population with chronic low back pain excluded, all this should be discussed in the discussion and be present in obtaining accurate conclusions consistent with reality.

Author Response

Congratulations to the authors for this research work, but the manuscript requires a number of important changes and corrections. The authors of the article set the following objective: effectiveness with the benefit of spa treatment for chronic LBP in people > 60 years old and to determine profiles of responders with trajectory model analysis.

The article requires an in-depth review of the English language by an expert.

Johanna Robertson made an in depth proof reading

My comments and suggested changes to the article are as follows:

1º The keywords should be a search and identification tool for your research work, look for the words that most closely resemble your manuscript, it would be advisable to use MesH terms, since you have not indicated for example the term hydrotherapy and balneotherapy.

We changed the key words : we added hydrotherapy and balneotherapy, and removed group-based multitrajectory modelling and changed elderly to aged regarding to the Pubmed Mesh

2º The title of the manuscript is unstructured and a bit confusing, rephrase the title, make it clear what type of study it is, is it a pilot study or an observational study, does it occur in patients with chronic pain or over 60 years old, can 30 year old patients have chronic low back pain?

We propose the following tittle : Identification of responders to balneotherapy among adults over 60 years of age with chronic low back pain: a pilot study with trajectory model analysis

It is a pilot study, concerning patients older than 60 years old with chronic low back pain. We also use trajectory model analysis which is the originality of the study

3º Indicate type of study in title and abstract.

Done

4º It should give a detailed and exhaustive description of the study population; incidence; classification; diagnosis; current treatments; physical, psychological, social and economic problems associated with the injury, according to the recommendations of the Guidelines for low back pain (an important aspect of the justification and importance of the study) both in the introduction and in the discussion.

It is difficult to give these elements for this study, since the population is quite specific (elderly subjects)

There are currently no recommendations for low back pain in the elderly, the recommendations concern common low back pain in subjects between 18 and 55 years of age

5º You use the following reference to justify in the text the use of balneotherapy "In 2009, in a systematic review of randomized controlled trials, Falagas et al. showed the effects of balneotherapy on pain in chronic LBP in 25 trials and a beneficial but statistically insignificant effect in 8 trials. The duration of effectiveness on pain ranged from 10 80 days to 1 year" this reference is more than 13 years old, there are more current studies such as Bai R, Li C, Xiao Y, Sharma M, Zhang F, Zhao Y. Effectiveness of spa therapy for patients with chronic low back pain: An updated systematic review and meta-analysis. Medicine (Baltimore). 2019 Sep;98(37):e17092. doi: 10.1097/MD.00000000000000017092. The introduction should be rewritten with more current bibliographic references, some are more than 10 years old, therefore the study presented is not justified with current evidence, which potentially limits the results obtained.

We replaced the reference with the paper by Bai et al. We have rewritten the introduction with more recent references.

6º The text is poorly written and gives rise to many doubts in the terms used, you refer to the treatment technique in different ways throughout the manuscript you call it Spa Treatment or Spa therapy or even allude to balneotherapy. What is spa therapy? In which similar populations has it been applied? What are the techniques that compose it? You should further develop this whole concept you are talking about and what has been done to date.

We replaced spa therapy by balneotherapy all along the paper.

We also gave a definition of balneotherapy.

7º The work deals with chronic low back pain and why can only people over 60 years of age participate? what is the basis for this? what clinical guide advises that it should only be from 60 years of age? why do they exclude a younger population? can a 30 year old person have chronic low back pain? It is important to justify this aspect in the introduction and that it be reflected in the limitations of the study (there is no section on the limitations of the study in the discussion).

The aim of the study was to assess the effect of spa therapy for older people because pharmacological treatment like NSAID’s are not indicated particularly for older adults. In addition, in France, the majority of low back pain patients who receive spa therapy are elderly

We have justified this in the limitations part of the study.

8º How were these participants diagnosed? Where were these participants diagnosed and referred from? Who performed the evaluation to determine that it was chronic low back pain?

The patients were referred by their physician, they were then seen by a specialist in the spa resort. The diagnosis was clinical. This wa added to the text.

Below are the inclusion and exclusion criteria:

“To be included in the study, patients had to be 60 to 80 years old, have mechanical lumbar spine pain for > 3 months. The criteria for non-inclusion; a history of spinal surgery; the presence of inflammatory rheumatism; a tumoral, traumatic or infectious lesion; the completion of a spa treatment in the 6 months before the start of the study; and an evolving herniated disc”

9º They should indicate the code provided by the ethics committee that approved this study with humans and that it is reflected that it complies with the declaration of Helsinki.

We added the code provided by the Ethics committee 2016/ CE 28 and that the research was conducted in accordance with the principles of declaration of Helsinki.

10º The sample size is very small. Why is the number of participants so small? How did you solve this problem? How was the sample size determined?

We thank the reviewer for this comment. Although the sample size may seem small, it is suitable and sufficient to guaranty satisfactory statistical power for the primary objective, i.e. to evaluate the effectiveness of conventional spa treatment in terms of effectiveness defined by a reduction in pain level. As described in the Statistics section, sample size estimation was based on Fleming's multi-stage design and calculations [Ratain MJ, Sargent DJ. Optimising the design of phase II oncology trials: the importance of randomisation. Eur J Cancer. 2009;45(2):275-280]. For the secondary objective, i.e. to determine the profile of a good responders to spa treatment with trajectory model analysis, it is interesting to observe that the statistical power can be also considered satisfactory. As reported in the submitted manuscript, after the group-based trajectory analysis, the change in pain level from D1 to D21 was -50% [95% CI -60; -27] for trajectory A and -13% [-33; 0] for trajectory B (p<0.001). The effect-size for this difference was 0.76 [0.30; 1.22]. The sample size to reach such a difference, for a two-tailed type I error at 5% and 85% statistical power, is 32 patients by group. So, with 32 and 46 patients, the statistical power was suitable to discuss this result.

However, we agree that further studies with larger sample sizes will be necessary to confirm the preliminary results from in this pilot study.

11º If the pathology under study is chronic, why was a long-term follow-up of the variables analyzed not performed?

The expected effect of the treatment was at 3 and 6 months, which is why we carried it out this way.

Moreover, it is very difficult to carry out a study with a follow-up of more than 6 months, which was not the purpose of this study.

12º They should insert the section on material and methods: study design, participants, outcomes, sample size, interventions (more detail on the techniques used, time spent in each of them and duration of the sessions, number of sessions per week). There were dropouts of participants, for reasons.

We have specified the study design: “This was a pilot, single-centre, prospective open cohort study, with repeated measurements using validated questionnaires” at the beginning of the materials and methods part.

The sample size is specified in the statistical paragraph : In view of recruitment capacity, feasibility and literature data, we chose a one-stage design with a lower bound of maximum inefficiency of about 50% and an upper bound of 65% (minimum efficiency), for risks of error of the alpha and beta set at 0.05 and 0.20, respectively (power of at least 80%). Under these assumptions, we needed to include 78 patients. The spa treatment was considered effective if ≥ 47/78 patients showed a reduction in pain level with an ES > 0.5.

We specified details for the intervention as follows : “the study took place at the Neyrac les Bains thermal baths over 4 months, All the treatments were all identical and pre-established for 6 of 7 days and lasted a total of 3 weeks. Four of the following hydrotherapy treatment were adminstrated each day : a thermal steam bath 45-48°C, cataplasm, aerobath with mud, general jet shower, mobiliza-tion pool, massage under thermal water, application of thermal water gel, exercise in hot thermal mud, and high pressure shower. The whole treatment lasted for 2 hours.

The composition of the water was as follows (per 1 L): silica 127.6 mg anions, H2CO3 1711.1 mg, SO4 16.6 mg, Cl 10.4 mg, cations, Ca++ 217 mg, Mg++ 63 mg, Na+ 274.4 mg, B 270 mg, Cu < 0.02 µg, Ph 6.45, Zn < 0.02 µg, Fe 15 mg, Se < 5 µg. “

We did not have any drop outs

13º It would be interesting to insert a flow diagram to better understand the process and images of the therapies used in the intervention.

We have not included a flow diagram as there were no drop outs.

14º Divide the sections of the results explaining each of the variables analyzed, the first part of the results are the characteristics of the participants.

We divided the results part in 4 paragraph 3.1. Patients, 3.2 Analysis of effectiveness of spa treatment on pain and secondary endpoints, 3.3 Effect on ODI, ASES and FABQ, 3.4 Trajectories of pain,

15º Discussions should cover the key findings of the study: discuss any previous research related to the topic to place the novelty of the discovery in the appropriate context, discuss possible shortcomings and limitations in its interpretations, discuss its integration into the current understanding of the problem and how. This advances current views, speculates on the future direction of research, and freely postulates theories that could be tested in the future, completed, and reformulated. The discussion should be rewritten to present serious errors.

There is no justification with the current literature of their results both for and against, as there are current studies of the use of balneotherapy, spa therapy in patients with chronic low back pain which have not been taken into consideration to discuss the results, I advise you to rewrite the discussion, then I attach a search where you have clinical trials and reviews on the topic of study that can be complementary to your discussion: https://pubmed.ncbi.nlm.nih.gov/?term=balneotherapy+and+low+back+pain&sort=date

We took in case the remark concerning the references and have updated many of them

1 Fullen B, Morlion B, Linton SJ, Roomes D, van Griensven J, Abraham L, Beck C,Wilhelm S, Constantinescu C, Perrot S. Management of chronic low back pain and the impact on patients' personal and professional lives: Results from an international patient survey. Pain Pract. 2022 ;22:463-477. doi:10.1111/papr.13103.

2 Knezevic NN, Candido KD, Vlaeyen JWS, Van Zundert J, Cohen SP. Low back pain. Lancet. 2021 ;398:78-92. doi: 10.1016/S0140-6736(21)00733-9.

6 Wong CK, Mak RY, Kwok TS, Tsang JS, Leung MY, Funabashi M, Macedo LG, Dennett L, Wong AY. Prevalence, Incidence, and Factors Associated With Non-Specific Chronic Low Back Pain in Community-Dwelling Older Adults Aged 60 Years and Older: A Systematic Review and Meta-Analysis. J Pain. 2022 ;23:509-534.doi: 10.1016/j.jpain.2021.07.012.

7 Bailly F, Trouvin AP, Bercier S, Dadoun S, Deneuville JP, Faguer R, Fassier JB, Koleck ML, Lassalle L, Le Vraux T, et al. Clinical guidelines and care pathway for management of low back pain with or without radicular pain. Joint Bone Spine. 2021 ;88:105227. doi:10.1016/j.jbspin.2021.105227.

8  Anderson DB, Shaheed CA. Medications for Treating Low Back Pain in Adults. Evidence for the Use of Paracetamol, Opioids, Nonsteroidal Anti-inflammatories, Muscle Relaxants, Antibiotics, and Antidepressants: An Overview for Musculoskeletal Clinicians. J Orthop Sports Phys Ther. 2022 ;52:425-431. doi: 10.2519/jospt.2022.10788.

9 Bhala N, Emberson J, A Merhi, S Abramson, N Arber, J A Baron,  C Bombardier, C Cannon, M E Farkouh, GA FitzGerald, et al. Vascular and upper gastrointestinal effects of non-steroidal anti-inflammatory drugs: meta-analyses of individual participant data from randomised trials. Lancet 2013; 382: 769–79

10 Monteiro C, Silvestre S, Duarte AP, Alves G. Safety of Non-Steroidal Anti-Inflammatory Drugs in the Elderly: An Analysis of Published Literature and Reports Sent to the Portuguese Pharmacovigilance System. Int J Environ Res Public Health. 2022 ;19:3541. doi: 10.3390/ijerph19063541.

13 Wong CK, Mak RY, Kwok TS, Tsang JS, Leung MY, Funabashi M, Macedo LG, Dennett L, Wong AY. Prevalence, Incidence, and Factors Associated With Non-Specific Chronic Low Back Pain in Community-Dwelling Older Adults Aged 60 Years and Older: A Systematic Review and Meta-Analysis. J Pain. 2022 ;23:509-534. doi: 10.1016/j.jpain.2021.07.012.

17 Oliveira CB, Maher CG, Pinto RZ, Traeger AC, Lin CC, Chenot JF, van Tulder M, Koes BW. Clinical practice guidelines for the management of non-specific low back pain in primary care: an updated overview. Eur Spine J. 2018;27:2791-2803. doi: 10.1007/s00586-018-5673-2.

18 Bai R, Li C, Xiao Y, Sharma M, Zhang F, Zhao Y. Effectiveness of spa therapy for patients with chronic low back pain: An updated systematic review and meta-analysis. Medicine (Baltimore). 2019 ;98:e17092. doi: 10.1097/MD.00000000000000017092.

19 Denis I, Fortin L. Development of a French-Canadian version of the Oswestry Disability Index: cross-cultural adaptation and validation. Spine (Phila Pa 1976). 2012 ;37:E439-44. doi: 10.1097/BRS.0b013e318233eaf9.

28 Ghrairi T, Chaftar N, Jarraud S, Berjeaud JM, Hani K, Frere J. Diversity of legionellae strains from Tunisian hot spring water. Res Microbiol. 2013;164:342-50. doi: 10.1016/j.resmic.2013.01.002.

We also added a limitations paragraph. The aim of the study was not to prove the efficacy of spa therapy but to define profile of good or bad responders, that we have done “Thus, individuals who seemed to be better responder present less functional impairment, are more frequent consumers of strong analgesics (stage II), have a higher ASES score and practice more physical activity than others

16º It presents very outdated and old references which do not correspond to the current evidence, reformulate its introduction and discussion according to the current literature and evidence, see references with more than 12 years of antiquity regarding the subject: [1,2,6,8, 9,10,13,14,15,16,17,19,21,24,26,27,28,32]

We agree, we have resumed the discussion by modifying the references as follow

1 Fullen B, Morlion B, Linton SJ, Roomes D, van Griensven J, Abraham L, Beck C,Wilhelm S, Constantinescu C, Perrot S. Management of chronic low back pain and the impact on patients' personal and professional lives: Results from an international patient survey. Pain Pract. 2022 ;22:463-477. doi:10.1111/papr.13103.

2 Knezevic NN, Candido KD, Vlaeyen JWS, Van Zundert J, Cohen SP. Low back pain. Lancet. 2021 ;398:78-92. doi: 10.1016/S0140-6736(21)00733-9.

6 Wong CK, Mak RY, Kwok TS, Tsang JS, Leung MY, Funabashi M, Macedo LG, Dennett L, Wong AY. Prevalence, Incidence, and Factors Associated With Non-Specific Chronic Low Back Pain in Community-Dwelling Older Adults Aged 60 Years and Older: A Systematic Review and Meta-Analysis. J Pain. 2022 ;23:509-534.doi: 10.1016/j.jpain.2021.07.012.

7 Bailly F, Trouvin AP, Bercier S, Dadoun S, Deneuville JP, Faguer R, Fassier JB, Koleck ML, Lassalle L, Le Vraux T, et al. Clinical guidelines and care pathway for management of low back pain with or without radicular pain. Joint Bone Spine. 2021 ;88:105227. doi:10.1016/j.jbspin.2021.105227.

8  Anderson DB, Shaheed CA. Medications for Treating Low Back Pain in Adults. Evidence for the Use of Paracetamol, Opioids, Nonsteroidal Anti-inflammatories, Muscle Relaxants, Antibiotics, and Antidepressants: An Overview for Musculoskeletal Clinicians. J Orthop Sports Phys Ther. 2022 ;52:425-431. doi: 10.2519/jospt.2022.10788.

9 Bhala N, Emberson J, A Merhi, S Abramson, N Arber, J A Baron,  C Bombardier, C Cannon, M E Farkouh, GA FitzGerald, et al. Vascular and upper gastrointestinal effects of non-steroidal anti-inflammatory drugs: meta-analyses of individual participant data from randomised trials. Lancet 2013; 382: 769–79

10 Monteiro C, Silvestre S, Duarte AP, Alves G. Safety of Non-Steroidal Anti-Inflammatory Drugs in the Elderly: An Analysis of Published Literature and Reports Sent to the Portuguese Pharmacovigilance System. Int J Environ Res Public Health. 2022 ;19:3541. doi: 10.3390/ijerph19063541.

13 Wong CK, Mak RY, Kwok TS, Tsang JS, Leung MY, Funabashi M, Macedo LG, Dennett L, Wong AY. Prevalence, Incidence, and Factors Associated With Non-Specific Chronic Low Back Pain in Community-Dwelling Older Adults Aged 60 Years and Older: A Systematic Review and Meta-Analysis. J Pain. 2022 ;23:509-534. doi: 10.1016/j.jpain.2021.07.012.

17 Oliveira CB, Maher CG, Pinto RZ, Traeger AC, Lin CC, Chenot JF, van Tulder M, Koes BW. Clinical practice guidelines for the management of non-specific low back pain in primary care: an updated overview. Eur Spine J. 2018;27:2791-2803. doi: 10.1007/s00586-018-5673-2.

18 Bai R, Li C, Xiao Y, Sharma M, Zhang F, Zhao Y. Effectiveness of spa therapy for patients with chronic low back pain: An updated systematic review and meta-analysis. Medicine (Baltimore). 2019 ;98:e17092. doi: 10.1097/MD.00000000000000017092.

19 Denis I, Fortin L. Development of a French-Canadian version of the Oswestry Disability Index: cross-cultural adaptation and validation. Spine (Phila Pa 1976). 2012 ;37:E439-44. doi: 10.1097/BRS.0b013e318233eaf9.

28 Ghrairi T, Chaftar N, Jarraud S, Berjeaud JM, Hani K, Frere J. Diversity of legionellae strains from Tunisian hot spring water. Res Microbiol. 2013;164:342-50. doi: 10.1016/j.resmic.2013.01.002.

17º The bibliography is not adapted to the standards of the journal.

The citations of the references in the text are erroneous, an in-depth revision of this aspect is needed, I recommend the authors to review the author's guide of the journal. Use the reference style, based on the American Chemical Society (ACS) style, both in the text and in the bibliography section, as indicated in the author's guide of the journal.

We modified the references in line with ACS style

18º You must reformulate your conclusions, referring only to the results obtained by you and mentioning the obtaining of conclusions with caution given the high number of limitations that this research project presents, such as the non-long-term follow-up of a chronic pathology, the sample size, no comparative group, the dispersion of treatment techniques, the selection bias of participants only older than 60 years, which leaves an important group of the population with chronic low back pain excluded, all this should be discussed in the discussion and be present in obtaining accurate conclusions consistent with reality.

This was a pilot study which explains that we did not take any control group

We agree that it could be interesting to get long term follow up but as many studies in spa therapy, we followed subjects till month 6

We had justified the sample size in the statistics part, we believe this is sufficient for the aim of our study

We selected only participants older than 60 years because it is the main aim of the study to show the benefit of spa therapy for people older than 60 years

Reviewer 3 Report

This study is a pilot testing of the effectiveness of spa treatment for patients with chronic LBP who are 60 years or older. It provides a trajectory analysis of the response, which is very useful for assessing improvement over the followup period.

In the abstract, the first sentence states that NSAIDs are the first line for treatment of chronic LBP. In the introduction as well, it is stated that these are the recommendations of the French health authorities. However, I have a strong reservation to stating this as a fact (that NSAIDs are first line treatment for chronic LBP) in the abstract and then again in the introduction.

I referred to these recommendations, and they state that NSAIDs should be used when risks and benefits are assessed for the shortest duration and with the least dose. If it is recommended for use "for the shortest duration" it can hardly be considered a treatment option that competes with spa treatment or physical therapy or other non-pharmacological modalities. Rather it complements them, "for the shortest duration possible".

Therefore, I would recommend NOT stating that spa treatment is an alternative for NSAIDs. Because this inherently means that they are both part of the same treatment categories, which they are not (one is pharmacological and one is a non-pharmacological), a distinction that is also made in the French recommendations. And because of the valid argument that NSAIDs are not safe in the elderly population under study.

Abstract: line 15, "the effectiveness of spa treatment". Please remove "with the benefits"

Abstract: please mention what trajectory A and B refer to. The abstract should be self-explanatory. One has to go all the way to the results section of the manuscript to know what each trajectory refers to.

Introduction: please mention what you mean by thermalism, and how is this similar or different from spa treatment.

Results: Page 5 line 200. The ODI score was significantly changed between time points, and so was the ASES: other symptoms. However, you state "We did not find any significant variation in ODI, ASES and FABQ over time (Table 3). Please explain.

Results: the description of the two pain trajectories is only explained well in the discussion section. It would be better to clarify that trajectory A refers to patients who had immediate pain reduction vs B who had less immediate improvement in the results section.

Discussion: Line 253, you mention a reference to the sentence "Our results show a benefit of spa treatment on pain". Please add what the reference provides (i.e agreement? disagreement?..etc)

Author Response

This study is a pilot testing of the effectiveness of spa treatment for patients with chronic LBP who are 60 years or older. It provides a trajectory analysis of the response, which is very useful for assessing improvement over the followup period.

In the abstract, the first sentence states that NSAIDs are the first line for treatment of chronic LBP. In the introduction as well, it is stated that these are the recommendations of the French health authorities. However, I have a strong reservation to stating this as a fact (that NSAIDs are first line treatment for chronic LBP) in the abstract and then again in the introduction.

I referred to these recommendations, and they state that NSAIDs should be used when risks and benefits are assessed for the shortest duration and with the least dose. If it is recommended for use "for the shortest duration" it can hardly be considered a treatment option that competes with spa treatment or physical therapy or other non-pharmacological modalities. Rather it complements them, "for the shortest duration possible".

Therefore, I would recommend NOT stating that spa treatment is an alternative for NSAIDs. Because this inherently means that they are both part of the same treatment categories, which they are not (one is pharmacological and one is a non-pharmacological), a distinction that is also made in the French recommendations. And because of the valid argument that NSAIDs are not safe in the elderly population under study.

We agree and have modified the abstract: because of the increased risk of adverse effects associated with the use of nonsteroidal anti-inflammatory drugs must be taken into account

And in the text we suppressed the following sentence “However, the therapy can be a relevant alternative to other therapies, particularly NSAIDs, in a population in which undesirable effects are increased and frequent.”

Abstract: line 15, "the effectiveness of spa treatment". Please remove "with the benefits"

Done

Abstract: please mention what trajectory A and B refer to. The abstract should be self-explanatory. One has to go all the way to the results section of the manuscript to know what each trajectory refers to.

We added effect size and  larger reduction in pain intensity for trajectory A, and smaller reduction in pain intensity for trajectory B

Introduction: please mention what you mean by thermalism, and how is this similar or different from spa treatment.

We suppressed thermalism and you can now find balneotherapy all along the article

Results: Page 5 line 200. The ODI score was significantly changed between time points, and so was the ASES: other symptoms. However, you state "We did not find any significant variation in ODI, ASES and FABQ over time (Table 3). Please explain. 

We apologize for this error, the changes have been made

Results: the description of the two pain trajectories is only explained well in the discussion section. It would be better to clarify that trajectory A refers to patients who had immediate pain reduction vs B who had less immediate improvement in the results section.

We modified the text as follows: “trajectory A (larger reduction in pain intensity) and in trajectory B (smaller reduction in pain intensity)”

Discussion: Line 253, you mention a reference to the sentence "Our results show a benefit of spa treatment on pain". Please add what the reference provides (i.e agreement? disagreement?..etc)

 We agree and add that our results agree with Karagülle et al

Round 2

Reviewer 2 Report

The authors have corrected some errors detected in the first revision of the manuscript, but they leave several of them without giving an answer or an adequate justification. I indicate the points that remain to be corrected and analyzed in depth in a subject as important as chronic low back pain and its affectation in elderly patients.

4º It should give a detailed and exhaustive description of the study population; incidence; classification; diagnosis; current treatments; physical, psychological, social and economic problems associated with the injury, according to the recommendations of the Guidelines for low back pain (an important aspect of the justification and importance of the study) both in the introduction and in the discussion.

It is difficult to give these elements for this study, since the population is quite specific (elderly subjects)

There are currently no recommendations for low back pain in the elderly, the recommendations concern common low back pain in subjects between 18 and 55 years of age

New answer: Chronic low back pain is the main cause of disability in the world. There are many references and guides on this subject, where all age ranges are discussed, where very relevant information on the pathology you are studying is indicated.

6º The text is poorly written and gives rise to many doubts in the terms used, you refer to the treatment technique in different ways throughout the manuscript you call it Spa Treatment or Spa therapy or even allude to balneotherapy. What is spa therapy? In which similar populations has it been applied? What are the techniques that compose it? You should further develop this whole concept you are talking about and what has been done to date.

We replaced spa therapy by balneotherapy all along the paper.

We also gave a definition of balneotherapy.

New answer: You continue without defining the concept of study in a clear way and with studies that justify this applications, you only indicate the following text which does not even have a bibliographic citation.

“Balneotherapy involves bathing in hot or 75 warm baths of natural mineral water in spa centers.”

7º The work deals with chronic low back pain and why can only people over 60 years of age participate? what is the basis for this? what clinical guide advises that it should only be from 60 years of age? why do they exclude a younger population? can a 30 year old person have chronic low back pain? It is important to justify this aspect in the introduction and that it be reflected in the limitations of the study (there is no section on the limitations of the study in the discussion).

The aim of the study was to assess the effect of spa therapy for older people because pharmacological treatment like NSAID’s are not indicated particularly for older adults. In addition, in France, the majority of low back pain patients who receive spa therapy are elderly

We have justified this in the limitations part of the study.

New answer: The limitations section of the study is not to justify the errors of the study. If you want to prove the effectiveness of balneotherapy in the elderly population with advanced age, you cannot justify this aspect with the fact that in France people over 55 mainly receive balneotherapy, again the term SPA appears, I think this concept should be clarified. In order to obtain reliable and valid conclusions and for this to be extrapolated to all persons with chronic low back pain over 60 years of age, there must be a more general vision of the problem, with literature that exists that justifies the importance and need to seek answers to this problem.

8º How were these participants diagnosed? Where were these participants diagnosed and referred from? Who performed the evaluation to determine that it was chronic low back pain?

The patients were referred by their physician, they were then seen by a specialist in the spa resort. The diagnosis was clinical. This wa added to the text.

New answer: We need clarification, a medical diagnosis based on which chronic low back pain guideline, tests, tests, teas; what did this clinical diagnosis consist of? how was it carried out?

11º If the pathology under study is chronic, why was a long-term follow-up of the variables analyzed not performed?

The expected effect of the treatment was at 3 and 6 months, which is why we carried it out this way.

Moreover, it is very difficult to carry out a study with a follow-up of more than 6 months, which was not the purpose of this study.

New answer: We continue to believe that this is a big mistake that increases the risk of bias since we are dealing with pathology of chronic origin and therefore to determine coherent and reliable results this follow-up should be carried out. This aspect, since it is insurmountable and causes the study to lose quality and evidence, should be discussed in the discussion, specified and clarified in the limitations and highlighted in the conclusions.

12º They should insert the section on material and methods: study design, participants, outcomes, sample size, interventions (more detail on the techniques used, time spent in each of them and duration of the sessions, number of sessions per week). There were dropouts of participants, for reasons.

Four of the following hydrotherapy treatment were adminstrated each day : a thermal steam bath 45-48°C, cataplasm, aerobath with mud, general jet shower, mobiliza-tion pool, massage under thermal water, application of thermal water gel, exercise in hot thermal mud, and high pressure shower.

New answer: The techniques used where they are described or justified do not appear in the introduction, nor in the methodology within the description of the interventions, nor are their application justified by previous studies, nor are they mentioned in the discussion.

13º It would be interesting to insert a flow diagram to better understand the process and images of the therapies used in the intervention.

We have not included a flow diagram as there were no drop outs.

New answer: The flowchart is a tool that allows the reader to make a mental diagram of the process performed, it has nothing to do with and influence dropouts, I recommend that you follow the CONSORT recommendations.

15º Discussions should cover the key findings of the study: discuss any previous research related to the topic to place the novelty of the discovery in the appropriate context, discuss possible shortcomings and limitations in its interpretations, discuss its integration into the current understanding of the problem and how. This advances current views, speculates on the future direction of research, and freely postulates theories that could be tested in the future, completed, and reformulated. The discussion should be rewritten to present serious errors.

New answer: The discussion is the poorest part of the paper and should be rewritten with a coherent and orderly structure, as indicated in the previous recommendations, it is not a matter of changing some bibliographic citations for others (not modifying the content) and the limitations are not well explained and defined.

16º It presents very outdated and old references which do not correspond to the current evidence, reformulate its introduction and discussion according to the current literature and evidence, see references with more than 12 years of antiquity regarding the subject: [1,2,6,8, 9,10,13,14,15,16,17,19,21,24,26,27,28,32]

New answer: When bibliographic citations are replaced by others, the content of the manuscript has to be modified, not a simple replacement, rewrite the content of the introduced citations since the content of the manuscript does not coincide with the content of the selected studies.

8º You must reformulate your conclusions, referring only to the results obtained by you and mentioning the obtaining of conclusions with caution given the high number of limitations that this research project presents, such as the non-long-term follow-up of a chronic pathology, the sample size, no comparative group, the dispersion of treatment techniques, the selection bias of participants only older than 60 years, which leaves an important group of the population with chronic low back pain excluded, all this should be discussed in the discussion and be present in obtaining accurate conclusions consistent with reality.

This was a pilot study which explains that we did not take any control group.

We agree that it could be interesting to get long term follow up but as many studies in spa therapy, we followed subjects till month 6.

We had justified the sample size in the statistics part, we believe this is sufficient for the aim of our study.

We selected only participants older than 60 years because it is the main aim of the study to show the benefit of spa therapy for people older than 60 years.

New answer: All this that you indicate to me are appreciations, you must solve the correction that I indicated previously and define the conclusions according to your manuscript.

Author Response

The authors have corrected some errors detected in the first revision of the manuscript, but they leave several of them without giving an answer or an adequate justification. I indicate the points that remain to be corrected and analyzed in depth in a subject as important as chronic low back pain and its affectation in elderly patients.

4º It should give a detailed and exhaustive description of the study population; incidence; classification; diagnosis; current treatments; physical, psychological, social and economic problems associated with the injury, according to the recommendations of the Guidelines for low back pain (an important aspect of the justification and importance of the study) both in the introduction and in the discussion.

It is difficult to give these elements for this study, since the population is quite specific (elderly subjects)

There are currently no recommendations for low back pain in the elderly, the recommendations concern common low back pain in subjects between 18 and 55 years of age

New answer: Chronic low back pain is the main cause of disability in the world. There are many references and guides on this subject, where all age ranges are discussed, where very relevant information on the pathology you are studying is indicated.

We made a review on pubmed using key words “chronic low back pain” with the limitation limitations “guideline, systematic review, ages + 65” for the last 10 years”. We did not find any guidelines for this population but we found few articles that can be cited

  • de Souza IMB, Sakaguchi TF, Yuan SLK, Matsutani LA, do Espírito-Santo AS, Pereira CAB, Marques AP. Prevalence of low back pain in the elderly population: systematic review. Clinics (Sao Paulo). 2019 Oct 28;74:e789. doi:10.6061/clinics/2019/e789.
  • Jones LD, Pandit H, Lavy C. Back pain in the elderly: a review. Maturitas. 2014 Aug;78(4):258-62. doi: 10.1016/j.maturitas.2014.05.004.
  • Kuss K, Becker A, Quint S, Leonhardt C. Activating therapy modalities in older individuals with chronic non-specific low back pain: a systematic review. Physiotherapy. 2015 Dec;101(4):310-8. doi: 10.1016/j.physio.2015.04.009.

Due to the lack of many studies on this topic, and the absence of any practice guidelines, it seems very interesting to conduct of pilot study on innovative stargegies

We add a paragraph in the introduction.

6º The text is poorly written and gives rise to many doubts in the terms used, you refer to the treatment technique in different ways throughout the manuscript you call it Spa Treatment or Spa therapy or even allude to balneotherapy. What is spa therapy? In which similar populations has it been applied? What are the techniques that compose it? You should further develop this whole concept you are talking about and what has been done to date.

We replaced spa therapy by balneotherapy all along the paper.

We also gave a definition of balneotherapy.

New answer: You continue without defining the concept of study in a clear way and with studies that justify this applications, you only indicate the following text which does not even have a bibliographic citation.

That is the definition gave by the Mesh of pubmed. We precise this in the text.

“Balneotherapy involves bathing in hot or 75 warm baths of natural mineral water in spa centers.”

As there is a lack of guidelines in the elderly for low back pain, hence the need for such a study

7º The work deals with chronic low back pain and why can only people over 60 years of age participate? what is the basis for this? what clinical guide advises that it should only be from 60 years of age? why do they exclude a younger population? can a 30 year old person have chronic low back pain? It is important to justify this aspect in the introduction and that it be reflected in the limitations of the study (there is no section on the limitations of the study in the discussion).

The aim of the study was to assess the effect of spa therapy for older people because pharmacological treatment like NSAID’s are not indicated particularly for older adults. In addition, in France, the majority of low back pain patients who receive spa therapy are elderly

We have justified this in the limitations part of the study.

New answer: The limitations section of the study is not to justify the errors of the study. If you want to prove the effectiveness of balneotherapy in the elderly population with advanced age, you cannot justify this aspect with the fact that in France people over 55 mainly receive balneotherapy, again the term SPA appears, I think this concept should be clarified. In order to obtain reliable and valid conclusions and for this to be extrapolated to all persons with chronic low back pain over 60 years of age, there must be a more general vision of the problem, with literature that exists that justifies the importance and need to seek answers to this problem.

We add a paragraph in the introduction with references concerning people over 60 with low back pain. Also, we rewrite the discussion and limitation.

8º How were these participants diagnosed? Where were these participants diagnosed and referred from? Who performed the evaluation to determine that it was chronic low back pain?

The patients were referred by their physician, they were then seen by a specialist in the spa resort. The diagnosis was clinical. This was added to the text.

New answer: We need clarification, a medical diagnosis based on which chronic low back pain guideline, tests, tests, teas; what did this clinical diagnosis consist of? how was it carried out?

The diagnosis was made on the HAS guidelines, and it is clinical. It was made by the GP of the patient and confirmed by the specialist seen in the spa centre.

11º If the pathology under study is chronic, why was a long-term follow-up of the variables analyzed not performed?

The expected effect of the treatment was at 3 and 6 months, which is why we carried it out this way.

Moreover, it is very difficult to carry out a study with a follow-up of more than 6 months, which was not the purpose of this study.

New answer: We continue to believe that this is a big mistake that increases the risk of bias since we are dealing with pathology of chronic origin and therefore to determine coherent and reliable results this follow-up should be carried out. This aspect, since it is insurmountable and causes the study to lose quality and evidence, should be discussed in the discussion, specified and clarified in the limitations and highlighted in the conclusions.

We agree with the reviewer and discussed this element both in the limitation and the conclusion of the study.

12º They should insert the section on material and methods: study design, participants, outcomes, sample size, interventions (more detail on the techniques used, time spent in each of them and duration of the sessions, number of sessions per week). There were dropouts of participants, for reasons.

Four of the following hydrotherapy treatment were adminstrated each day : a thermal steam bath 45-48°C, cataplasm, aerobath with mud, general jet shower, mobiliza-tion pool, massage under thermal water, application of thermal water gel, exercise in hot thermal mud, and high pressure shower.

New answer: The techniques used where they are described or justified do not appear in the introduction, nor in the methodology within the description of the interventions, nor are their application justified by previous studies, nor are they mentioned in the discussion.

We precised the intervention in the methodology with references. We do not think that is right to precise it in the introduction. We mention it in the discussion.

13º It would be interesting to insert a flow diagram to better understand the process and images of the therapies used in the intervention.

We have not included a flow diagram as there were no drop outs.

New answer: The flowchart is a tool that allows the reader to make a mental diagram of the process performed, it has nothing to do with and influence dropouts, I recommend that you follow the CONSORT recommendations.

We add a flow chart

15º Discussions should cover the key findings of the study: discuss any previous research related to the topic to place the novelty of the discovery in the appropriate context, discuss possible shortcomings and limitations in its interpretations, discuss its integration into the current understanding of the problem and how. This advances current views, speculates on the future direction of research, and freely postulates theories that could be tested in the future, completed, and reformulated. The discussion should be rewritten to present serious errors.

New answer: The discussion is the poorest part of the paper and should be rewritten with a coherent and orderly structure, as indicated in the previous recommendations, it is not a matter of changing some bibliographic citations for others (not modifying the content) and the limitations are not well explained and defined.

We rewrite a part of the discussion and explain more the limitations

16º It presents very outdated and old references which do not correspond to the current evidence, reformulate its introduction and discussion according to the current literature and evidence, see references with more than 12 years of antiquity regarding the subject: [1,2,6,8, 9,10,13,14,15,16,17,19,21,24,26,27,28,32]

New answer: When bibliographic citations are replaced by others, the content of the manuscript has to be modified, not a simple replacement, rewrite the content of the introduced citations since the content of the manuscript does not coincide with the content of the selected studies.

We do not agree with the reviewer as we up-dated the references and took the time to verify their content

8º You must reformulate your conclusions, referring only to the results obtained by you and mentioning the obtaining of conclusions with caution given the high number of limitations that this research project presents, such as the non-long-term follow-up of a chronic pathology, the sample size, no comparative group, the dispersion of treatment techniques, the selection bias of participants only older than 60 years, which leaves an important group of the population with chronic low back pain excluded, all this should be discussed in the discussion and be present in obtaining accurate conclusions consistent with reality.

This was a pilot study which explains that we did not take any control group.

We agree that it could be interesting to get long term follow up but as many studies in spa therapy, we followed subjects till month 6.

We had justified the sample size in the statistics part, we believe this is sufficient for the aim of our study.

We selected only participants older than 60 years because it is the main aim of the study to show the benefit of spa therapy for people older than 60 years.

New answer: All this that you indicate to me are appreciations, you must solve the correction that I indicated previously and define the conclusions according to your manuscript.

The conclusion has been modified

Reviewer 3 Report

Thank you, I have no further comments.

Author Response

We warmly thank reviewer 3 for the job done